# Optical Isomerization and Photo-Patternable Properties of GeO₂/Ormosils Organic-Inorganic Composite Films Doped with Azobenzene

**Xuehua Zhang [1],\*, Qian Wang [1], Shun Liu [1], Wei Zhang [1], Fangren Hu [1,2],\* and Yongjin Wang [2]**

1   College of Electronic and Optical Engineering & College of Microelectronics, Nanjing University of Posts and Telecommunications, Nanjing 210023, China; 15702941917@163.com (Q.W.); liushun15050046679@163.com (S.L.); chanway@njupt.edu.cn (W.Z.)
2   Peter Grunberg Research Center, Nanjing 210023, China; wangyj@njupt.edu.cn
\*   Correspondence: zhangxuehua@njupt.edu.cn (X.Z.); hufr@njupt.edu.cn (F.H.); Tel.: +86-25-8586-6402 (F.H.)

**Abstract:** GeO₂/organically modified silane (ormosils) organic-inorganic composite films containing azobenzene were prepared by combining sol-gel technology and spin coating method. Optical waveguide properties including the refractive index and thickness of the composite films were characterized by using a prism coupling instrument. Surface morphology and photochemical properties of the composite films were investigated by atomic force microscope and Fourier transform infrared spectrometer. Results indicate that the composite films have smooth and neat surface, and excellent optical waveguide performance. Photo-isomerization properties of the composite films were studied by using a UV–Vis spectrophotometer. Optical switching performance of the composite films was also studied under the alternating exposure of 365 nm ultraviolet light and 410 nm visible light. Finally, strip waveguides and microlens arrays were built in the composite films through a UV soft imprint technique. Based on the above results, we believe that the prepared composite films are promising candidates for micro-nano optics and photonic applications, which would allow directly integrating the optical data storage and optical switching devices onto a single chip.

**Keywords:** composite films; azobenzene; sol-gel; photo isomerization; optical switching





## 1. Introduction

Optical switches play an increasingly important role in all-optical networks [1,2]. Azobenzene and its derivatives are one of the most widely used optical switches [3,4]. Photosensitivity is the core of optical switch [5]. Reversible molecular optical switch azobenzene is a compound with photochromic properties [6]. Photochromism refers to the photoinduced transformation of a compound between two (or more) forms with different absorption spectra and thus different colors [7]. Azobenzene can undergo cis-trans reversible isomerization under ultraviolet light and visible light, leading to significant changes in molecular geometry and end-to-end distance, which means that azobenzene can be used as an optical switch [8]. In the natural state, azobenzene is in a stable trans state because of its low intrinsic energy. Under the irradiation of ultraviolet light, the trans isomer is excited and immediately transformed into cis isomer. Under visible light or heated conditions, cis isomers are transformed into trans isomers [9]. Therefore, the photoisomerization of azobenzene makes it a promising optical switch in the fields of chemical engineering, biomedicine and optoelectronics [10–12]. Photoisomerization based on azobenzene also makes polymers containing azobenzene molecules have photo-isomerization characteristics. These photosensitive materials do not need to be directly contacted and can be remotely controlled, thus avoiding the contamination of the materials [13].

Organic-inorganic composite materials have both the advantages of inorganic oxides and organic materials, such as the film-forming property and process-ability of organic polymers, and the high hardness and high temperature resistance of inorganic materials [14–16].

Moreover, compared with organic polymer materials, their optical properties can be greatly improved [17]. Therefore, organic-inorganic composites have been widely studied as photoresponsive materials [18,19]. There are many methods to prepare organic-inorganic composite materials, such as sol-gel technology, blending method, molecular self-assembly method and so on [20]. Among them, sol-gel process is based on hydrolysis and condensation, which has high product purity, small film crack, simple operation and strong adhesion, and can be widely used in different materials [21,22].

In our previous work, titanium-based composite films doped with azobenzene were been studied [23]. We know germanosilicate glasses are widely used as the core materials for the production of optical fibers due to their high optical transmittance in the visible and infrared range. Additionally, Bragg gratings and second harmonic generation can be formed due to the ultra-violet photosensitivity of germanosilicate glasses [14]. All the above properties provide a significant potential for photonic applications such as data storage and information processing.

In this paper, germanium dioxide based organic-inorganic composite thin films doped with azobenzene are prepared by combining sol-gel technology and spin-coating method. The optical and photo-isomerization properties of the composite films are characterized. In addition, photosensitive groups are also incorporated into the material, which means that strip waveguides and microlens arrays can be easily built on the composite thin films by using a UV soft imprint technology. The composite materials prepared in this paper are expected to become ideal materials in the optics applications.

## 2. Materials and Methods

### 2.1. Film Preparations

The composite films were prepared by combining a low temperature sol-gel technique and spin coating method. To avoid the reaction between germanium alkoxides and moisture in the air, the preparation of the composite sol was carried out in a dry nitrogen environment. The sol was prepared by three solutions. Solution I: 1 mole of γ-glycidoxypropyltrimethoxysilane (GLYMO) was mixed with 4 moles of ethanol and 4 moles of deionized water, and after being stirred for 30 min, 0.01 mole of hydrochloric acid (HCl, 37 wt. %) was added and stirred for another 1 h until the solution became homogeneous. Solution II: 3-methacryloxypropyltrimethoxysilane (MEMO), isopropanol and deionized water were mixed at a molar ratio of 1:3:3, then 0.01 mol of 37 wt. % HCl was added as the catalyst, and the mixed solution was stirred for about an hour. Solution III: 1 mole of germanium isopropoxide ($GeO_2$) was mixed with 4 moles of acetylacetone and the solution was agitated an hour. After that, the three solutions were mixed at a molar ratio of $GeO_2$: GLYMO: MEMO with 1:2:2 and stirred for about 2 h. Then the azobenzene powder was added with different weight ratios of 1 wt. %, 3 wt. % and 5 wt. %, respectively. The final mixture was stirred for about 25–30 h at room temperature, following the composite film was obtained by a spin coating method. It should be noted that, to effectuate the photo-polymerization of the carbon-carbon double bond from MEMO under UV light irradiation, a photo-initiator in the form of bis (2,4,6-trimethylphenyl) phosphine with 4 wt. % was added into the final mixture before spin coating process. A 0.22 μm pore filter was attached to the syringe to remove the foreign particles. Silicon, silica and glasses were used as the substrates, and they were respectively cleaned with acetone, ethanol and deionized water for about 10 min and dried in pure nitrogen. One layer of the composite sol-gel film was deposited onto the substrate at a spinning speed of 3500 rpm for 35 s. Finally, the film samples were heated in an oven at different temperatures of 100, 200, 300, 400 and 500 °C for 10 min. The films were then irradiated under UV light at a wavelength of 365 nm for different times from 5 s to 30 min to study the photo-isomerization properties.

### 2.2. Patterned Film by UV Imprint Technique

In this paper, strip waveguides with a waveguide width of 20 μm and microlens arrays with a microlen diameter of 20 μm were respectively fabricated on the films. There

are mainly three steps. Firstly, the photoresist masters with strip waveguides and microlens arrays were fabricated by combining a photo-lithography technique and a photoresist thermal reflow method, which the detailed process has been described in our previous work [24]. Secondly, the PDMS (polydimethylsiloxane) soft template was replicated from the photoresist mater. The PDMS precursor and its curing agent are poured into a paper cup at a weight ratio of 10:1, and after fully stirring, they are allowed to stand for 15 min. After the air bubbles were completely removed, the mixed PDMS colloid solution is poured onto the photoresist microlens arrays or strip waveguides masters, and then the whole sample is put into a vacuum drier at 90 °C for 1 h for curing. After the PDMS is completely cured, it is gently removed from the photoresist masters and obtained the PDMS soft templates. Thirdly, one thick layer of the composite film was spin-coated on the glass substrate at a spinning speed of 900 rpm for 60 s. The prepared PDMS soft template is imprinted in the film with a pressure of 10,000 Pa. After standing in air for 10 min, the whole sample was cured in a drying oven at 90 °C for 60 min for the evaporation of water and organic solvents. Then the sample was exposed under UV light for 30 min with a PLS-SXE xenon lamp (Beijing Bofeilai Technology Co., Ltd., Beijing, China). After full solidification, the PDMS soft template is peeled off, and obtained the patterned composite films.

### 2.3. Sample Characterizations

Different techniques have been used to characterize the properties of the composite films. Surface morphology and roughness of the composite films were observed by AFM (Digital Instruments Nanoscope IIIa, Santa Barbara, CA, USA) by means of a tapping mode. Optical propagation modes, the refractive index and thickness of the composite films were investigated by a prism coupler (Metricon 2010, USA). Photo-isomerization and optical switching properties of the composite films deposited on glass substrates were measured by a UV-Vis spectrophotometer (UNICO UV-2800A, Shanghai, China) in the range of 200–600 nm. The TGA of the composite gel powder was measured by a Perkin Elmer STA 8000, USA. Photochemical properties of the composite films deposited on silicon substrates were characterized by a FTIR (Perkin Elmer Two, USA, with a resolution of $\pm 1 cm^{-1}$, no heated ART attached) spectrometer in the range of 4000–400 $cm^{-1}$. PLS-SXE xenon lamp was used as the ultraviolet and visible light source, which the emitted light wavelength could be adjusted by changing the filter. The surface morphology of the films and the microstructures were observed by SEM (Hitachi S-4800, Japan), operated at an accelerating voltage of 15 kV.

## 3. Results and Discussion

### 3.1. Surface Topography Characteristics

The surface morphology properties of the composite films doped with 5 wt. % azobenzene were studied. Figure 1 shows the AFM images of the composite films deposited on silicon substrates and baked at different temperatures of (a) 50, (b) 100, (c) 200, (d) 300, (e) 400 and (f) 500 °C. It can be seen from Figure 1 that the films heated below 200 °C have a relatively smooth surface, while the film surfaces baked above 200 °C are a little rough. Additionally, the root mean square (RMS) roughness of the composite films was also measured, and the RMS roughness value is below 0.8 nm for all the samples, in the scanning range of 5 μm × 5 μm, which is small enough to meet the requirements of optical elements.

To further study the surface morphology properties, the composite films were observed by SEM; the results are shown in Figure 2. It should be mentioned that the composite films were deposited on silicon substrates and heated respectively at a low temperature of 50 °C in Figure 2a,b and a high temperature of 200 °C in Figure 2c,d. It can be easily found that, the composite films heated at 50 °C have a relatively smooth surface, while the surface of thin films heated at 200 °C are a little porous, which the results are constant with the AFM images in Figure 1.

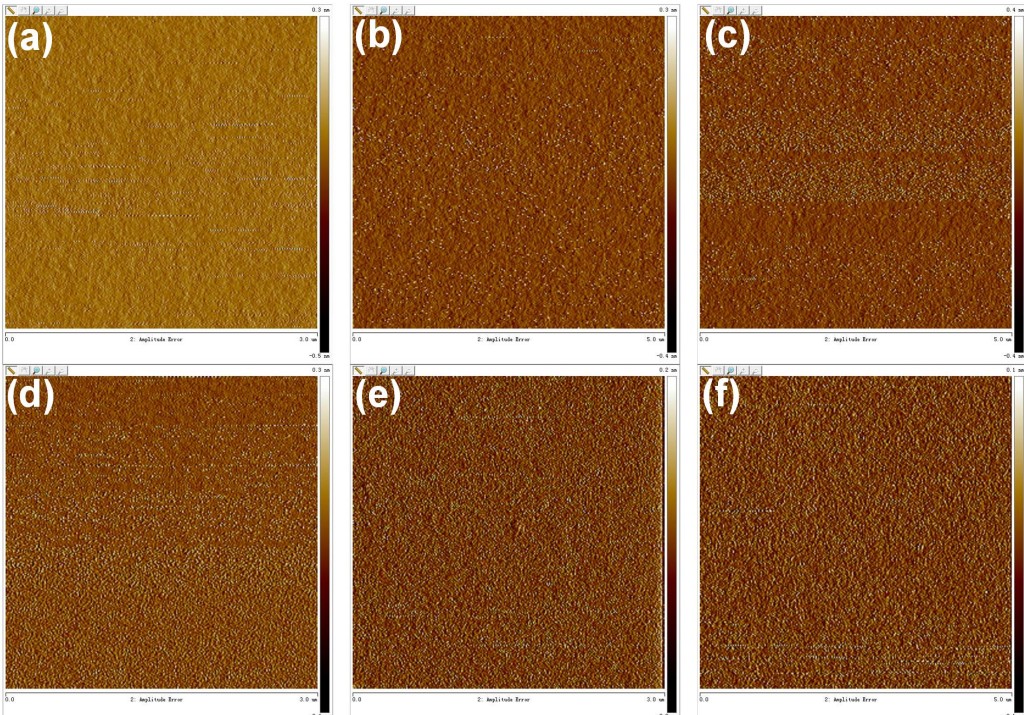

**Figure 1.** AFM images of the composite films containing 5 wt. % azobenzene and heated at different temperatures of (**a**) 50, (**b**) 100, (**c**) 200, (**d**) 300, (**e**) 400 and (**f**) 500 °C.

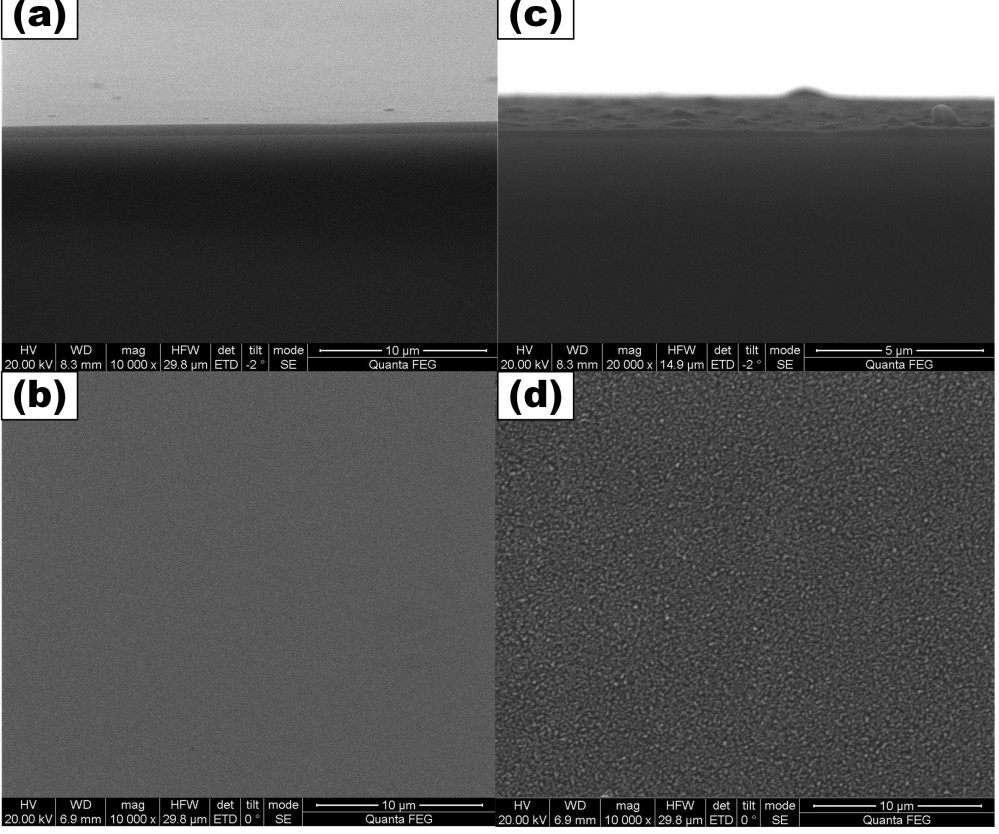

**Figure 2.** SEM images of the composite films containing 5 wt. % azobenzene and heated at (**a,b**) 50 °C and (**c,d**) 200 °C.

### 3.2. Optical Waveguide Characteristics

Figure 3 shows the TE transmission modes of the composite film doped with 5 wt. % azobenzene, and at the wavelength of 633 nm. It should be mentioned that the film was deposited on silica substrate and heated at 200 °C for 10 min. It can be found in Figure 2 that there are four wave troughs in total. However, only two modes of $TE_0$ and $TE_1$ can be considered as the guided modes of the composite films. It is because of that, the refractive index of the substrate is about 1.46, and only two refractive indexes calculated from the $TE_0$ and $TE_1$ are higher than this value, and the other two wave troughs represent the substrate modes. Additionally, we also found that three TE modes can be detected from the films heated below 100 °C, and it is difficult for the TE modes to be detected from the films heated above 200 °C, which can be explained by the results of AFM images in Figure 1, that the films become a little porous when the heat treatment temperature is above 200 °C with the decomposition and combustion of organic groups.

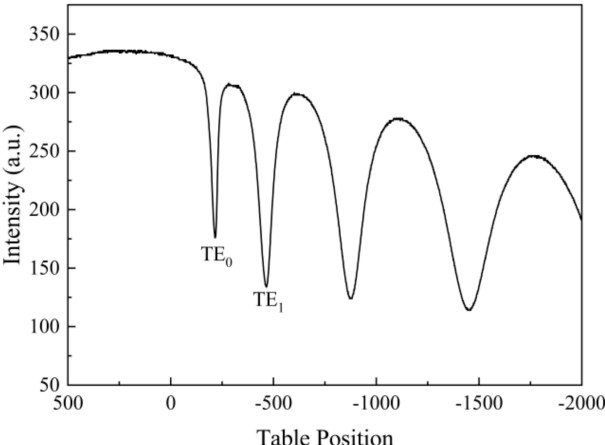

**Figure 3.** Optical guided TE modes of the composite film containing 5 wt. % azobenzene and heated at 200 °C.

Figure 4 shows the refractive index and thickness of the composite films with 5 wt. % azobenzene content and heated at different temperatures. It can be easily seen in Figure 4 that, with increasing the heat treatment temperature from 50 to 500 °C, the film thickness gradually decreases. When the heat treatment temperature is below 200 °C, the film thickness decreases almost linearly, which is due to the evaporation of water and organic solvents and the decomposition of some organic groups. As the heat treatment temperature rises from 200 to 300 °C, the film thickness decreases greatly, which is caused by the decomposition and combustion of a large number of organic groups. Further increasing the heat treatment temperature to 500 °C, the film thickness was almost unchanged, indicating that the films are inorganic dense [25]. It can also be found in Figure 4 that, as the heat treatment temperature rises, the refractive index of the films decreases first and then increases. As mentioned above, the composite film is dense below 100 °C due to the full filling of organic solvents and organic groups among the pores of inorganic oxide chains. As the heat treatment temperature rises to 200 °C, on the one hand, the film becomes porous due to the evaporation of water and organic solvents and the combustion of some organic groups like azobenzene, on the other hand, the film becomes much denser due to the heat-polymerization progress [26]. Thus, it probably induces some refractive index decrease in this temperature range. When the heat treatment temperature rises to 300 °C, the increase of refractive index is mainly caused by the further combustion and decomposition of organic groups in the system. With further increasing the heat treatment temperature to 500 °C, the film becomes inorganic dense, and the refractive index changes a little, which can be explained by the TGA results in Figure 5.

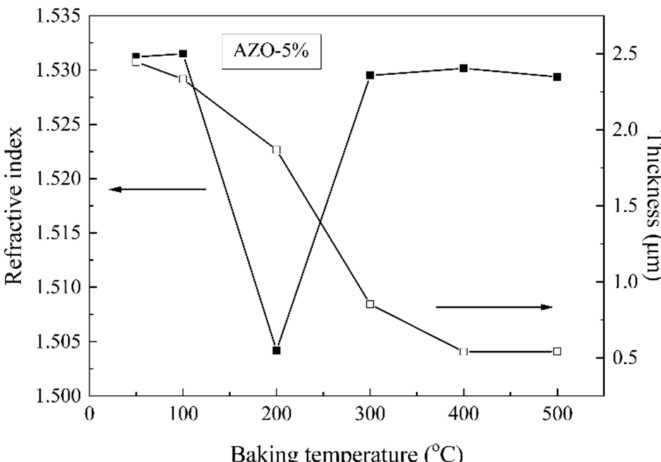

**Figure 4.** The refractive index and thickness of the composite films containing 5 wt. % azobenzene and heated at different temperatures.

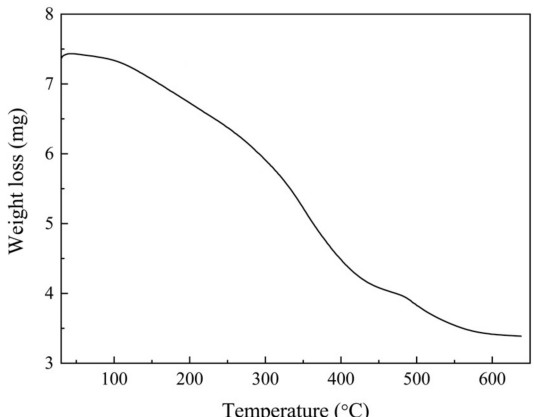

**Figure 5.** TGA curve of the gel powder of the composite material doped with 5 wt. % azobenzene.

### 3.3. Structural Characteristics

Figure 5 shows the TGA curve of the gel powder of the composite material doped with 5 wt. % azobenzene. The composite sol material was dried in air at room temperature for two weeks and obtained the gel powder. Then about 7.4 mg of the gel powder was used for the TGA test. It can be seen from Figure 5 that there are mainly three weight loss stages, namely, below 300 °C, between 300 and 450 °C and from 450 to 600 °C. There is about 20% weight loss in the first weight loss stage below 300 °C, which is mainly caused due to the evaporation of water and the thermal decomposition and volatilization of organic solvents. It can be observed that the weight loss changes rapidly between 300 to 450 °C, and the weight loss is about 24% in this weight loss stage, which is attributed to the carbonization and the combustion of some organic compounds. In the last weight loss stage, the weight loss is ascribed to the further combustion of organic compounds. There is almost no weight loss above 600 °C, indicating that the organic groups have been completely burnt off and the sample should be inorganic glass. The AFM results in Figure 1 and the optical transmission modes in Figure 3 can also be explained by the TGA results.

Figure 6 shows the effects of heat treatment temperature on the photochemical properties of the composite films. The composite films were doped with 5 wt. % azobenzene and deposited on silicon substrates. It can be seen in Figure 6 that all the curves have an obvious peak near 1085 cm$^{-1}$, which belongs to Si-O-R tensile vibration of ethoxy group directly bonded with silicon [27]. With increasing the heat treatment temperature, the strength of this band decreases. The small peaks observed at about 610 cm$^{-1}$ for all samples come

from the silicon substrates. Two small peaks at 1296 cm$^{-1}$ and 975 cm$^{-1}$ on both sides of the main peak band correspond to -CH$_3$ rocking vibration from Si-O-CH$_3$ functional groups and Ge-O-Ge anti-symmetric stretching [28], and the two peaks begin to weaken at 300 °C and disappear completely at 400 °C. It is worth noting that the peak at 2905 cm$^{-1}$ caused by -CH$_2$- symmetric stretching almost disappear when the heat treatment temperature reaches 400 °C. In addition, two peaks can be clearly observed at about 1639 cm$^{-1}$ and 1719 cm$^{-1}$, which are attributed to vinyl group C=C stretching mode and the carbonlyl ester groups, respectively [29]. Both disappear as the heat treatment temperature reaches 300 °C.

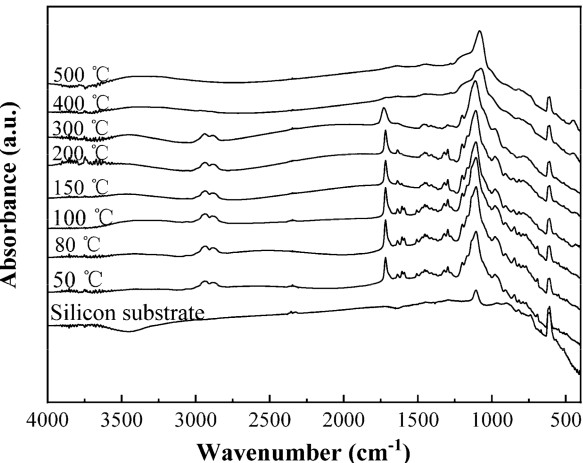

**Figure 6.** FTIR absorption spectra of the composite films containing 5 wt. % azobenzene and heated at different temperatures.

### 3.4. Photo-Isomerization Properties

Figure 7 shows the photo-isomerization properties of the composite films. It should be mentioned that the composite films were doped with 1 wt. % azobenzene, deposited on glass substrates and heated at 50 °C for 10 min. The composite films were firstly irradiated under ultraviolet light of 365 nm for different times from 0 s to 30 min, following the absorption spectra of the composite films was measured immediately. The whole experiment was conducted in a dark room. It can be seen in Figure 7 that there is a strong absorption peak at 350 nm, which corresponds to the π-π* electronic transition form in the side chain of trans-azobenzene. A weak absorption peak appears at the wavelength of 450 nm, which corresponds to the n-π* electronic transition form in the side chain of cis-azobenzene. Azobenzene is in a stable trans isomer at room temperature in the dark, and when irradiated by un-polarized ultraviolet light of 365 nm, the azobenzene chromophore undergoes photo-isomerization from trans-isomers to cis-isomers. Thus, it can be seen from Figure 7 that, with increasing UV irradiation time from 0 s to 20 min, the peak intensity at 350 nm decreases, and the peak intensity at 450 nm accordingly increases. We also find that by further increasing the ultraviolet exposure time from 20 to 30 min, the intensity of the two peaks changes a little, indicating that the trans-isomer and the cis-isomer reach an equilibrium state and coexist in the film system.

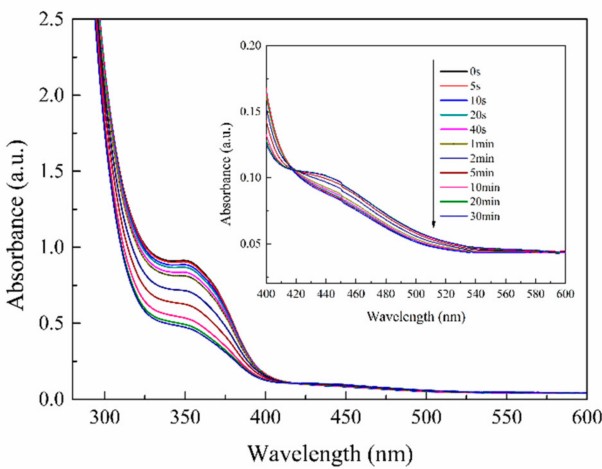

**Figure 7.** UV–Vis absorption changes of the composite films irradiated by UV light of 365 nm for different times from 0 s~30 min.

### 3.5. Optical Switch Performance

Figure 8 shows the optical switching properties of the composite films. The composite films were exposed under an un-polarized UV light of 365 nm and visible light of 450 nm alternately for five cycles at room temperature in the dark. It should be mentioned that the composite film was doped with 1 wt. % azobenzene content, deposited on glass substrate and heated at 200 °C for 10 min, the irradiance power of UV light and visible light respectively is 16 mW/cm$^2$ and 12 mW/cm$^2$. The detailed process is that the composite film was irradiated under UV light for 1 min, and the UV-Vis absorption spectrum of the film was measured immediately. Subsequently, the film was irradiated under visible light for 1 min, following the UV-Vis absorption spectrum the film was measured again. Five such cycles were repeated, and after that the absorption value at the wavelength of 350 nm in each UV–Vis absorption spectrum was carefully recorded, and the optical switching characteristics were obtained, as shown in the diagram in Figure 8. It can be seen from Figure 8 that there are several periods of reversible photo-isomerization under the alternate irradiation of ultraviolet light and visible light. When the composite film is irradiated with 365 nm ultraviolet light for 1 min, the absorption value of the film at the wavelength of 350 nm is about 0.42, which is equivalent to the "on" state of the switch. Similarly, when the composite film is irradiated with 450 nm visible light for 1 min, the absorption value of the film at wavelength of 350 nm is about 0.50, which is equivalent to the "off" state of the switch [30].

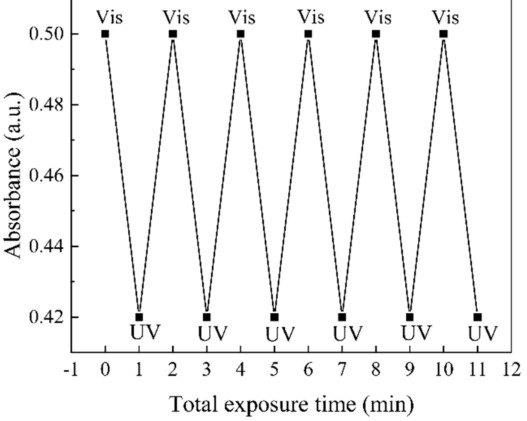

**Figure 8.** Optical switching characteristics of the composite films by alternating UV and visible light irradiation for five cycles (light intensity, UV: 16 mW/cm$^2$, vis: 12 mW/cm$^2$).

### 3.6. Surface Morphology of the Patterned Film

Photosensitive organic groups have been incorporated into the composite films, which makes that the composite films are sensitive to UV light, thus micro-nano arrays structure can be easily prepared on the film by using UV nanoimprint technology. Figure 9 shows SEM images of the patterned composite films with different microstructures of (a) strip waveguides and (b) microlens arrays. It can be seen from Figure 9 that the patterned films have good uniformity, smooth surface, and no redundant impurities. We believe that the prepared composite films in our work will have great application prospect in micro-nano optics and photonic applications.

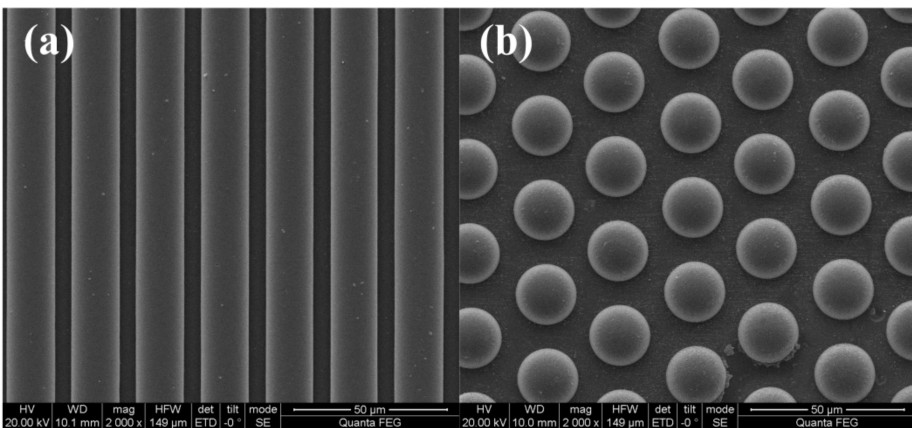

**Figure 9.** SEM images of the patterned composite films with different microstructures of (**a**) strip waveguides and (**b**) microlens arrays.

## 4. Conclusions

In this paper, $GeO_2$/ormosils organic-inorganic composite thin films doped with azobenzene were prepared by combining sol-gel technology and spin coating method. The surface morphology results show that the composite films have a smooth and neat surface, and the surface roughness value of the films is about 0.4 nm, which is small enough for the optical elements applications. The optical guided TE modes of the composite films were measured by using a prism coupling instrument at 633 nm, results show that there is a relatively good optical propagation properties for the composite films heated below 200 °C. Influence of heat treatment temperature on the refractive index and thickness of the composite films were studied. Results show that as the heat treatment temperature rises, the film thickness decreases, and the refractive index of the films first decreases slowly and then increases. Photo-isomerization and optical switching characteristics of the films were studied. Results show that the composite films have good photo-isomerization properties. Additionally, there are several periods of reversible photo-isomerization under the alternate irradiation of ultraviolet light and visible light, indicating a good optical switching performance and strong fatigue. Finally, the composite films were patterned respectively with strip waveguides and microlens arrays, by using a UV imprint technology. The prepared composite films have important application value in micro-nano optics areas.

**Author Contributions:** Writing—review and editing, X.Z.; writing—original draft preparation, Q.W.; formal analysis, S.L.; project administration, W.Z.; funding acquisition, F.H. and Y.W. All authors have read and agreed to the published version of the manuscript.

**Funding:** National Natural Science Foundation of China (NSFC) (Nos: 61605086, 51602160, 61574080, 61274121).

**Institutional Review Board Statement:** Not applicable.

**Informed Consent Statement:** Not applicable.

**Data Availability Statement:** Not applicable.

**Acknowledgments:** The authors would like to thank Xinwen Zhang for the experimental assistance.

**Conflicts of Interest:** The authors declare no conflict of interest.

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
