# Peer review of "Optical Isomerization and Photo-Patternable Properties of GeO2/Ormosils Organic-Inorganic Composite Films Doped with Azobenzene"

_coatings, doi:10.3390/coatings11070818_

Round 1

Reviewer 1 Report

The article presents GeO2/organically modified silane composite films containing azobenzene, starting from the preparation methods and continuing with their optical waveguide properties together with surface morphology and photochemical properties. The investigation is accurate and methodology well presented, while the results obtained are important in view of the applications envisaged.

However, some ambiguities need to be clarified:

Chapter 2.2. - sample characterization (lines 94-104)

  1. If thin films were deposited on glass substrates how the analysis was possible in the ultraviolet range?
  2. Data on experimental procedure must be completed (FTIR - no of scans, resolution, device used for heating, is it a heated ATR ?.....) Please clarify!

Chapter 3.2. - Optical waveguide characteristics (lines 133-153)

In order to obtain data on the thermal stability of composites and effects obtained during heating, it is necessary to supplement data with thermal analysis (thermogravimetry and Dynamic scanning calorimetry), otherwise they are only assumptions based on literature data, not on your own determinations;

Moreover, before conclusions, investigations on the effect of thickness gradient on optical guiding losses and eventually dispersion as a function of temperature of waveguides and substrates should be studied.

Overall, the article will be ready for publishing after solving these minor issues.

Reviewer 2 Report

The present work by Zhang et al. describes the synthesis and characterization of organic-inorganic composite films doped with azobenzene. In my opinion the presentation of the results is rather unclear, the novelty of the study and the motivation of some experiments is not explained properly. I cannot recommend this work for publication in the present form, however it can be reconsidered after some significant changes are done.

The main problem of this study is unclear presentation of the results. The authors prepared the films of different composition on different substrates and annealed the as-prepared films at different temperatures. However, there is no clear comparison in terms of chemical composition, substrate or processing conditions. For instance, Fig. 1 demonstrates only AFM image of the film containing 5% of azobenzene processed at 100 C on Si. Figure 2 shows the results for the film processed on SiO2 at 200 C. Figure 5 already shows the results for the film produced on glass at 50 C and containing only 1% of azobenzene. The experiments performed with different samples, there is no clear connection between them.

The novelty and motivation of the present study must be underlined in the introduction. It is not clear what new insight this work brings to the field. What is the reason of using germanium oxide? It is not explained.

Line 79. the mixture was stirred for 30 hours. Why so long?

How many layers were deposited? It is not mentioned in the experimental part.

The measurement mode is not indicated for AFM.

Line 112. RMS value is given as 0.450. Are the authors sure that this value can be given so precisely? 0.001 nm.

Lines 126-127. The porosity of the films cannot be determined by FTIR. Please, provide SEM images.

Lines 152-153. There is no evidence of the formation of dense inorganic films. Please, provide SEM images and XRD patterns.

Figure 3. How many films did the authors test? RI value for 200 C sample looks like an accident mistake.

Figure 4. Please, provide FTIR spectrum of the non-coated substrate.

Figure 5 is unclear. Every color should be linked to the time value. In the present form it looks opposite compared to the related text.

Section 3.6. This part sounds like experimental details. Maybe it should be transferred to the experimental section.

What is the aim of annealing at high temperature, if the authors do not use annealed (300-500C) coatings for further experiments?

Round 2

Reviewer 2 Report

accept